# An improved framework to develop Personas applied to willingness of vaccination against COVID-19 in the general population

Emanuele Tauro
*Department of Cardiollogic Researches*
*Istituto Auxologico Italiano*
Milan, Italy
e.tauro@auxologico.it

Alessandra Gorini
*Department of Clinical Sciences and*
*Community Health*
*University of Milan*
Milan, Italy
alessandra.gorini@unimi.it

Enrico Gianluca Caiani
*Dept. of Electronics, Information and*
*Bioengineering (DEIB)*
*Politecnico di Milano*
Milan, Italy
enrico.caiani@polimi.it

*Abstract*— **Vaccine hesitancy is characterized by a multitude of different sociodemographic and psychological factors that require interventions and information to be tailored to the specific users. Thus, the aim of this work is to develop an improved framework to create Personas to identify the characteristics of the population willing to be vaccinated, to facilitate the development of tailored eHealth-based interventions to increase vaccine uptake. Data was collected through an online survey at the beginning of 2021. Multiple dimensionality reduction methods were used to create Personas using K-medoids clustering with PAM algorithm and agglomerative hierarchical clustering with average linkage. The optimal number of Personas and dimensionality reduction methods were chosen through the evaluation of average silhouette graph, total within sum of square distances and percentage of statistically different attributes between clusters. From 1070 respondents, three Personas were identified: one (Persona 3) represented the least willing to be vaccinated compared to the other two (P < 0.001). This information was highly and significantly correlated with lower trust in institutions (P < 0.001), lower level of education (P < 0.001) and lower fear of COVID-19 pandemic (P < 0.001) when compared to the other two Personas. An improved version of a framework to create Personas was applied to identify the characteristics of the population that was less willing to be vaccinated. This approach used a novel indicator, representing the percentage of statistically different attributes among clusters, to identify the optimal number of Personas and the most proper preprocessing methods. Results suggested that tailored interventions should focus on taking advantage of closer social circle of vaccine-hesitant individuals to rebuild trust. This study is the first to use Personas to evaluate willingness of vaccination against the COVID-19 pandemic in the general population to identify potential tailored solutions.**

*Keywords— eHealth, Personas, Persuasive System Design, Vaccination, Behavioral Change*

## I. Introduction (*Heading 1*)

The 27th of December 2020 marked the beginning of the vaccination campaign against COVID-19 in Italy and Europe [1]. While vaccines have demonstrated efficacy in reducing mortality and morbidity associated with the pandemic, a notable portion of the general population exhibited hesitancy towards vaccination [2]. The World Health Organization (WHO) defines vaccine hesitancy as the delay in acceptance or outright refusal of vaccines despite their availability [3], identifying it as a growing concern in the European Region [4].

Individuals base their decision to vaccinate on factors such as personal risk perception, attitudes, social and cultural norms, habits, and other influential elements [5]. Additionally, the assimilation of vaccination information is filtered through individual experiences and knowledge, influenced by characteristics like age, gender, education, and socioeconomic status [4]. This diversity underscored the necessity for adopting various communication channels and strategies to effectively convey information to diverse demographic groups. The attainment of herd immunity emerged as a central objective for safeguarding against the pandemic [6]. Consequently, understanding the personal characteristics, reasons, and needs of those exhibiting hesitancy towards vaccination became crucial for the development of tailored interventions and messages aimed at addressing the concerns of hesitant individuals.

Personas, fictional representations of archetypes of real-world people, are commonly used to understand the needs and requirements of a target population [7], and are becoming increasingly used in healthcare to perform personalization of interventions [8], [9], [10], [11]. We hypothesized that an approach based on Personas' creation could be applied to assess the characteristics of the population that is willing, or not, to be vaccinated.

Thus, the aim of this study is the definition of an improved version of a framework, present in current literature [9], applied to the development of Personas to assess the characteristics of the population that is willing, or not, to be vaccinated. These Personas will aid in understanding how to tailor engagement activities, interventions and messages aimed at increasing the percentage of population willing to be vaccinated in possible future pandemic scenarios.

## II. Methods

### A. Survey definition and Data collection

Data were collected through a web survey developed by a team of domain experts, and disseminated to the general population using the Qualtrics® platform in the months of January and February 2021, corresponding to the beginning of the COVID-19 vaccination campaign in Italy [1], [12].

The survey was composed of five blocks of questions, including both validated and ad-hoc questionnaires, each assessing a different kind of information from the respondents.

This study was funded in part by Italian Ministry of Health (Bando Ricerca Finalizzata 2021 – "InTake Care Trial" RF-2021-12374708, C.U.P. E43C22000980001).

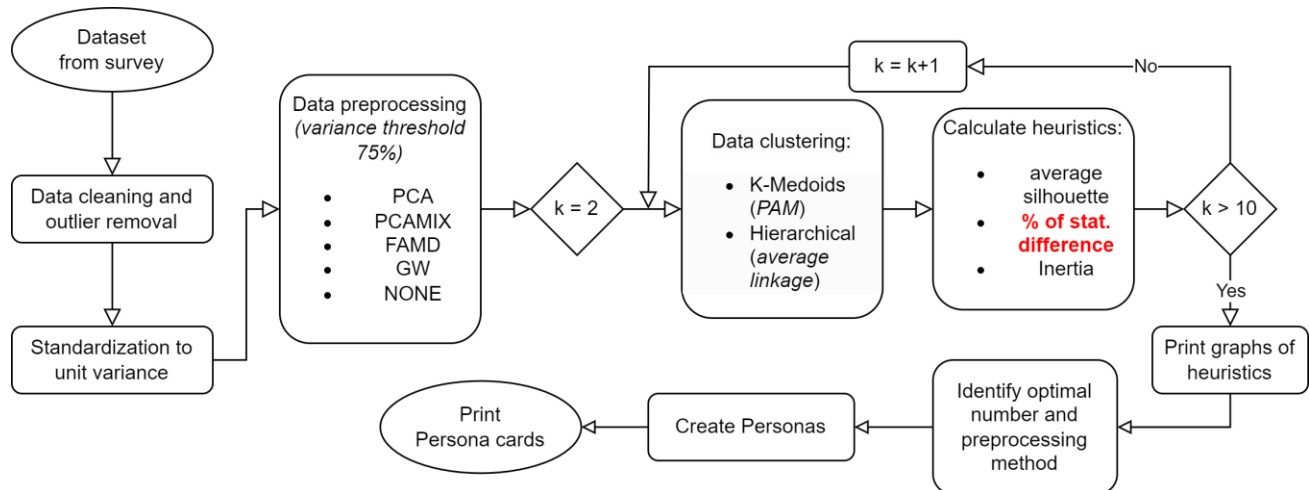

Fig. 1. Flowchart representing the proposed data processing. The beginning and ending point are shown in ellipses. The different processes are shown in rounded rectangles, while the starting and ending point of the loop are shown in diamond shapes. The value k represents the varying number of clusters, ranging from 2 to 10, over which the clustering and calculation of evaluation parameters are performed.

A complete description of the survey can be found in the work by Giuliani et al. [12].

In the first block (n = 6), sociodemographic factors were collected, focusing on age, gender, marital status and education. The second group (n = 17) included information about the physical and psychological status of the respondent, such as their perceived health situation, their physical and psychological status and if they were following psychological therapy. Furthermore, the Generalized Anxiety Disorder Questionnaire (GAD-7) [13] was used to assess the perceived anxiety of the respondent, while the Multidimensional Health Locus of Control Scale (MHLCS) was used to assess the respondent's belief of being in control of their own health [14]. The third group of questions (n = 13) was relevant to the respondent's reaction to COVID-19. The fourth group (n = 9) focused on beliefs about vaccines and willingness of vaccination. Finally, the fifth group (n = 3) focused on the trust in governmental, healthcare and scientific institutions. The survey included a total of 48 questions, comprising both quantitative and categorical variables.

The study was approved by the Ethical Committees of the University of Milan (approval number: 16/21; 16 February 2021). The respondents gave their explicit electronic consent to data treatment and usage, in accordance with the rules defined by the GDPR, with obtained data anonymized by removing possible identifiable personal data such as the IP.

### B. Data Analysis

The method used to analyze the dataset and develop Personas is derived from current literature [9], with modifications to adapt it to the current specific analysis. Fig. 1 shows the flowchart with all the steps performed to analyze the data.

Starting from the survey dataset, respondents who did not initially consent with their data usage or did not complete the whole questionnaire were removed from further analysis. Questions that referred to the same questionnaire (such as GAD-7 or MHLCS) or covered similar topics were combined into single values. This process resulted in 29 features, derived from the 48 questions, that were used for subsequent analysis.

All data were standardized to unit variance using the interquartile range, to improve robustness towards outliers [15]. Subsequently, the following preprocessing methods were performed independently and compared against each other, and against a dataset without preprocessing (NONE): 1) Principal Component Analysis (PCA) [16], usable after one-hot-encoding of categorical variables; 2) Principal Component Analysis of Mixed Data (PCAMIX) [17], a variation of PCA able to distinguish between numerical and categorical data; 3) Factor Analysis of Mixed Data (FAMD) [18], able to distinguish between numerical and categorical data and analyze questions divided into groups separately; 4) Gower's distance (GW) [19], identifying the distance between data points on a mixed dataset. For all preprocessing methods, the variance threshold was identified at 75% of the total variance explained by the dataset.

Resulting data were clustered through the usage of K-Medoids clustering with the Partitioning Around Medoids (PAM) algorithm [20], using the K-means++ initialization method [21] and the square Euclidean distance to evaluate the distance between data points. Datasets resulting from Gower's distance preprocessing were also clustered by hierarchical agglomerative clustering, using precomputed distances and median linkage [22].

K-Medoids required the number of clusters K to be decided a priori before clustering, as no golden standard to define the optimal number is currently available in scientific literature for K-Medoids or hierarchical clustering. Accordingly, for each preprocessing method, clustering was iteratively performed varying the number of clusters K from 2 to 10.

To determine the optimal preprocessing method and the optimal number of clusters, three scores were calculated: the total-within sum of square distances (or inertia), the average silhouette score, and the percentage of statistically different attributes.

The total-within sum of square distances evaluates the distance between points in the cluster and their respective medoids, with the optimal value being identified through the use of the elbow criterion [23]. For clusters obtained with GW preprocessing and hierarchical analysis, the inertia was

calculated after identifying the medoid as the element of the cluster that minimizes the distance from all other elements.

The average silhouette provides an evaluation of clustering validity [24] and results in values between -1 and +1, with higher values indicating more separation among clusters. Within the Persona development approach, the focus on the average silhouette is placed exclusively on having a value above 0.

The percentage of statistically different attributes is a proposed novel indicator that performs statistical analysis after clustering for every variable in the dataset, by comparing data among clusters in a pairwise manner. Nonparametric tests were performed, as the data distribution was deemed not normal. For quantitative variables, Kruskal-Wallis test [25] was performed, followed by pairwise Mann-Whitney U tests [26]. For qualitative variables, pairwise Fisher's tests were applied [27] and calculated performing a Monte Carlo simulation to take into account high computational costs [28]. In order to adjust the resulting p-values for multiple tests, Bonferroni correction [29] was used for $2 \leqslant K \leqslant 4$ clusters. For $K \geqslant 5$, the Benjamini-Hochberg correction [30] was preferred, as Bonferroni correction was deemed too conservative for high number of tests [31]. The final score was then calculated as the number of tests resulting in statistically significant differences over the total number of performed tests, ranging from 0 (no statistical difference) to 1 (all performed tests showed statistical significance). Using these scores, it was possible to identify the optimal number of clusters, and thus Personas, to be developed.

In the percentage of statistically different attributes graph, higher values indicate better separation between the obtained Personas.

*C. Personification*

The result of the proposed framework is a Persona Table, a tabular representation of the chosen set of Personas that contains all the analyzed attributes for each Persona. Nominal categorical variables displayed the mode and percentage of respondents, while quantitative variables showed the median value with its 25th and 75th percentiles. Fisher's exact test determined the p-value for nominal categorical variables, while Kruskal-Wallis test was used for quantitative variables. The table included symbols indicating significant pairwise tests.

Persona cards can be developed from the table, with the aim of making visually immediate representations of Personas [32], making them feel real to increase their efficacy [7]. Accordingly, a name was chosen, and a face picture was generated by a Generative Adversarial Network [33]. Furthermore, a short textual description was included to summarize most of the characteristics of the Persona.

Domain experts identified trust in institutions, anxiety (represented through the GAD-7 index) and fear of COVID-19 as the goals of the Personas (i.e., the attributes most relevant to willingness of vaccinations), which were represented through circular indicators representing low, middle and high levels of the respective attribute. A traffic light-based color coding was introduced: for anxiety and fear of COVID-19, low values were identified with green, moderate with yellow and high with red. For trust in institutions, low levels were coded in red, high levels in green, while moderate levels in yellow. These levels were assigned

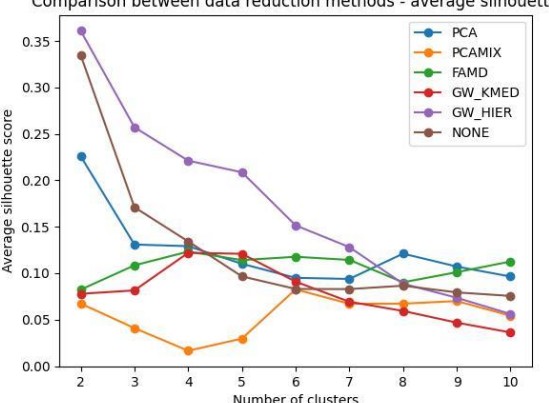

Fig. 2. Graph of the average silhouette score computed over a varying number of clusters ranging from 2 to 10 for the applied preprocessing methods (PCA, PCAMIX, FAMD, GW) as well as for no preprocessing (NONE). Clustering was performed with K-Medoids using PAM algorithm for all methods except for GW_HIER, obtained through hierarchical agglomerative clustering. All scores are higher than zero, suggesting that all preprocessing methods and number of clusters are usable.

based on statistical difference between Personas. For trust in institutions and fear of COVID-19 the one presenting the lowest score was set to low while the one with the highest score was set to high. For anxiety, the high level was set only for Personas with median values above the threshold of 9 points, otherwise a moderate (yellow) anxiety was set for scores ranging from 5 to 9, and a low (green) anxiety for scores of 4 or lower, as described in literature [13], [34]. The willingness to vaccinate was represented by a binary red (no) or green (yes) indicator.

Other interesting attributes, related to sociodemographic factors, were identified through tags. All other attributes were considered in the definition of the biography of the Persona, providing additional context and information.

### III. RESULTS

A total of 1101 participants responded to the online survey during the months of January and February 2021. After removing 31 respondents who did not consent or did not complete the survey, the final sample consisted of 1070 respondents. Females (n = 722, 67.5%) were significantly more prevalent (p < 0.001) than males (n = 348, 32.5%), and had a median age (25th; 75th percentile) of 42 (30; 54), while males had a median age of 46 (33; 59) and were significantly older (p < 0.001) than females. Among the population, the vast majority chose to be vaccinated (n = 913, 85.33%) versus those that avoided vaccination (n = 157, 14.67%).

Fig. 2 shows the graph of the average silhouette values for number of clusters ranging from 2 to 10, using K-Medoids Clustering with PAM algorithm for all preprocessing methods, with hierarchical clustering applied only on Gower's distance. The highest silhouette value was found for GW_HIER with K = 2 clusters, while the lowest was obtained with PCAMIX and K = 4 clusters. The heuristic was positive for all methods and potential number of clusters, suggesting that any combination could be used.

In Fig. 3, the graph representing the percentage of statistically different attributes is shown. PCAMIX showed the best results for $K \leqslant 4$, and thus was identified as the optimal preprocessing method. The percentage of statistically

different attributes in PCAMIX was the highest for K=2, with K=3 showing the second highest value. The lowest value was identified for GW_HIER and K = 10.

In Fig. 4, the graphs representing the total within sum of square differences are shown. For the purpose of clarity, each preprocessing method is displayed separately. This graph does not compare methods but evaluates the optimal number of clusters for each method individually. Thus, for clarity purposes, only the PCAMIX subgraph is observed. In the inertia PCAMIX subgraph presented in Fig. 4, an elbow is identified for K=3. Although the elbow is difficult to identify graphically, at K=3 the graph presents the maximum difference in inertia scores between the preceding and succeeding number of clusters.

Taking all three heuristics into account it was identified, with additional information from domain experts to discern between K=2 and K=3, that optimal decision was to consider three Personas with the PCAMIX preprocessing method.

In Table 1, the Personas Table derived from the application of the proposed framework to willingness of vaccination to COVID-19 in the general population is presented. The sex distribution showed a majority of women in all Personas, with Persona 2 presenting the highest percentage of men. However, as no difference was found in characteristics between men and women, Personas were created identical for both genders. No respondent identified as non-binary. The three Personas were unbalanced, with Persona 1 (n = 454) representing those with medium fear of COVID-19, medium trust in institutions, medium anxiety and medium perceived control over their health. Persona 2 (n = 388) represented those that had the highest fear of COVID-19, lowest anxiety, highest trust in institution, and highest perceived control over their health. Persona 3 (n = 228) represented those that had the lowest fear of COVID-19, highest anxiety, lowest trust in institution and lowest perceived control over their health. Regarding age, Personas 2 and 3 were significantly older than Persona 1. Personas 1 and 2 had a degree, while Persona 3 completed only high school. All Personas deem themselves healthy and tended to have jobs not in healthcare sector, with a higher concentration of healthcare jobs in Persona 1. Persona 3 presented the lowest fear of COVID-19 and related outcome for themselves, or their family and friends. Persona 2 showed the highest fear, as well as the highest perceived probability of contracting it. Persona 3 had the lowest value in trusting institutions, and in utility of vaccines. It was also not interested in recommending vaccination to others, the least interested in preventive behaviors and showed severe doubts about the health situation in Italy one year after the beginning of the vaccination program. GAD-7 scores showed the highest values of anxiety for Persona 1. Furthermore, Persona 2 showed the lowest values in MHLCS, identifying higher perceived control over health. Finally, Persona 3 showed the highest percentage of people that did not want to be vaccinated, with 50% of its respondents that decided to refuse vaccination.

In Fig. 5, the developed Persona Cards are presented, each representing a member of the corresponding cluster. As no statistical differences were found between genders, one Persona Card was created for each cluster, representing both the male and female version, modifying only the names and photos. Silvia and Marco, in cluster 1, have a moderate value

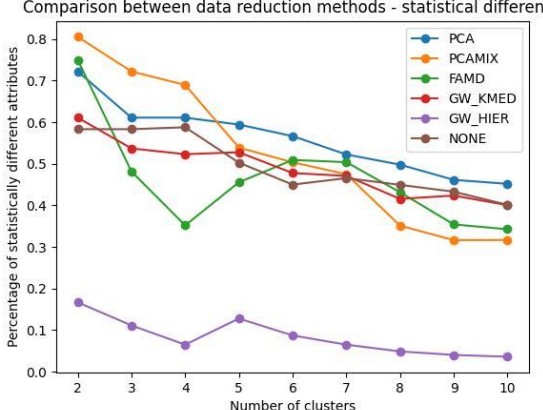

Fig. 3. Graph of the percentage of statistically different attributes computed over a varying number of clusters ranging from 2 to 10 for the applied preprocessing methods (PCA, PCAMIX, FAMD, GW), as well as for no preprocessing (NONE). Clustering was performed with K-Medoids using PAM algorithm for all methods except GW_HIER, obtained through hierarchical agglomerative clustering

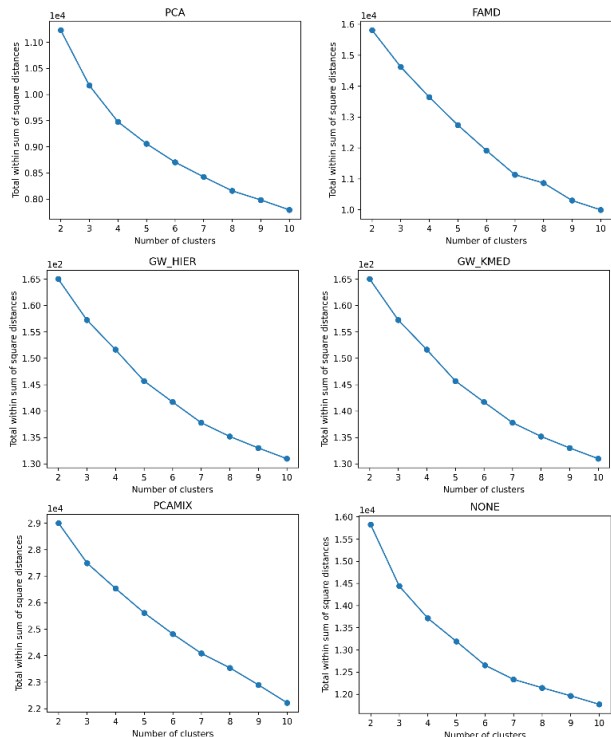

Fig. 4. Graph of the total within sum of square distances computed over a varying number of clusters ranging from 2 to 10 for the applied preprocessing methods (PCA, PCAMIX, FAMD, GW), as well as for no preprocessing (NONE). Clustering was performed with K-Medoids using PAM algorithm for all methods except GW_HIER, obtained through hierarchical agglomerative clustering.

of anxiety, moderate trust in institutions and fear of COVID-19, and they are willing to be vaccinated. Barbara and Attilio, in cluster 2, have low values of anxiety and high trust in institutions, but also high fear of COVID-19, being willing to be vaccinated. Elisa and Franco, in cluster 3, present low levels in all three indexes: anxiety, fear of COVID-19 and trust in institutions. Furthermore, they are not willing to be vaccinated.

TABLE I. COMPARISON BETWEEN THE THREE PERSONAS RELATED TO THE DATASET OF WILLINGNESS OF VACCINATION AGAINST COVID-19 IN THE GENERAL POPULATION, SHOWING ONLY THE ATTRIBUTES WITH STATISTICALLY SIGNIFICANT DIFFERENCE. VALUES ARE REPORTED AS MEDIAN (25TH; 75TH) FOR CONTINUOUS VARIABLES, AND MODE (%) FOR BINARY AND NOMINAL VARIABLES

| Variable | Persona 1 (n = 454) | Persona 2 (n = 388) | Persona 3 (n = 228) | P value |
|---|---|---|---|---|
| Age | 38.0 (29.0; 51.0) | 44.0 (34.0; 57.0) * | 49.0 (34.5; 58.0) * | < 0.001 |
| Sex | F (79%) | F (52%) * | F (71%) & | < 0.001 |
| Civil status | married (59%) | married (72%) * | married (74%) * | < 0.001 |
| Education | degree (44%) | degree (40%) | high school (45%) * & | < 0.001 |
| Job in Healthcare | No (62%) | No (72%) * | No (71%) * | 0.002 |
| Healthy participant | Yes (73%) | Yes (65%) * | Yes (71%) | 0.027 |
| Psychological status | 4.0 (3.0; 4.0) | 4.0 (4.0; 4.0) * | 4.0 (4.0; 4.0) * | < 0.001 |
| Follow psychological therapy | Yes (65%) | No (73%) * | No (61%) * & | < 0.001 |
| Perceived COVID severity self | 5.0 (4.0 ; 6.0) | 5.0 (4.0 ; 7.0) * | 5.0 (3.0 ; 5.25) & | < 0.001 |
| Perceived COVID severity familiy | 3.0 (2.0; 4.0) | 0.0 (0.0; 1.0) * | 1.0 (0.0; 3.0) * & | < 0.001 |
| Probability of contracting COVID self | 5.0 (4.0; 6.0) | 5.0 (4.0; 7.0) * | 5.0 (3.0; 5.6) * & | < 0.001 |
| Probability of contracting COVID similar others | 5.0 (4.0; 7.0) | 6.0 (5.0; 7.0) * | 5.0 (4.0; 6.0) * & | < 0.001 |
| COVID health damage | 3.0 (3.0; 4.0) | 4.0 (3.0; 4.0) * | 3.0 (3.0; 4.0) & | < 0.001 |
| Is COVID more severe than flu? | 4.0 (4.0; 5.0) | 5.0 (4.0; 5.0) * | 4.0 (4.0; 4.0) * & | < 0.001 |
| Fear of going to hospital without COVID | 2.0 (2.0; 3.0) | 2.0 (2.0; 3.0) | 3.0 (2.0; 3.0) * & | < 0.001 |
| Fear of contracting COVID Self | 3.0 (3.0; 4.0) | 4.0 (3.0; 4.0) * | 3.0 (2.0; 4.0) * & | < 0.001 |
| Fear of contracting COVID Family | 4.0 (4.0; 5.0) | 4.0 (4.0; 5.0) * | 4.0 (3.0; 4.0) * & | < 0.001 |
| Fear of contracting COVID Friends | 4.0 (3.0; 4.0) | 4.0 (4.0; 4.0) * | 3.0 (3.0; 4.0) * & | < 0.001 |
| Trust in institutions | 12.0 (10.0; 13.0) | 12.0 (11.0; 13.0) * | 10.0 (8.0; 11.0) * & | < 0.001 |
| COVID Vaccine utility | 4.0 (4.0; 5.0) | 5.0 (4.0; 5.0) | 4.0 (3.0; 4.0) * & | < 0.001 |
| What would change your mind about vaccines | indications by authorities (76%) | indications by authorities (80%) * | not changing mind (43%) * & | < 0.001 |
| Do you recommend COVID vaccine others | Yes (95%) | Yes (93%) | Yes (52%) * & | < 0.001 |
| Continue prevention behaviors after COVID vaccine | Yes (99%) | Yes (98%) | Yes (95%) * | 0.001 |
| Acquaintances with different vaccine opinion | Yes, some (75%) | Yes, some (63%) * | Yes, some (71%) * & | < 0.001 |
| Better sanitary situation after 1year of vaccines | Yes (88%) | Yes (88%) * | don't know (51%) * & | < 0.001 |
| GAD7 Total score | 6.0 (4.0; 10.0) | 5.0 (3.0; 7.0) * | 5.0 (2.0; 8.0) * | < 0.001 |
| MHLCS Total score | 14.0 (11.0; 17.0) | 12.0 (9.0; 16.0) * | 14.0 (11.0; 18.0) & | < 0.001 |
| Willingness being vaccinated | Yes (95%) | Yes (94%) | Yes (50%) * & | < 0.001 |

## IV. DISCUSSION

In this study, an improved approach based on Personas' creation was applied to assess the characteristics of the population that is willing, or not, to be vaccinated, to support future development of tailored interventions to increase vaccine uptake.

Results showed the capability of this framework to compare multiple preprocessing methods on the same data and use heuristics to identify the optimal number of Personas and the optimal preprocessing method to reduce data dimensionality and increase the efficacy of clustering for Personas. This study introduces a methodological and an applicational novelty when compared to similar studies within the field of Personas creation for healthcare.

### A. Methodological novelty: percentage of statistically different attributes graph

The key methodological novelty of this study is the introduction of a novel metric in the Persona development process: the percentage of statistically different attributes. This metric represents a significant advancement when compared to previous works based on quantitative Persons creation [9], [10], [11]. While previous approaches primarily relied exclusively on average silhouette and inertia graphs, together with previous knowledge from domain experts, the proposed novel metric provides a more comprehensive and quantitative basis that can be used, in addition to already existing metrics, to assess the optimal preprocessing method and number of Personas to be developed. It is based on the premise that effective Personas require statistically different

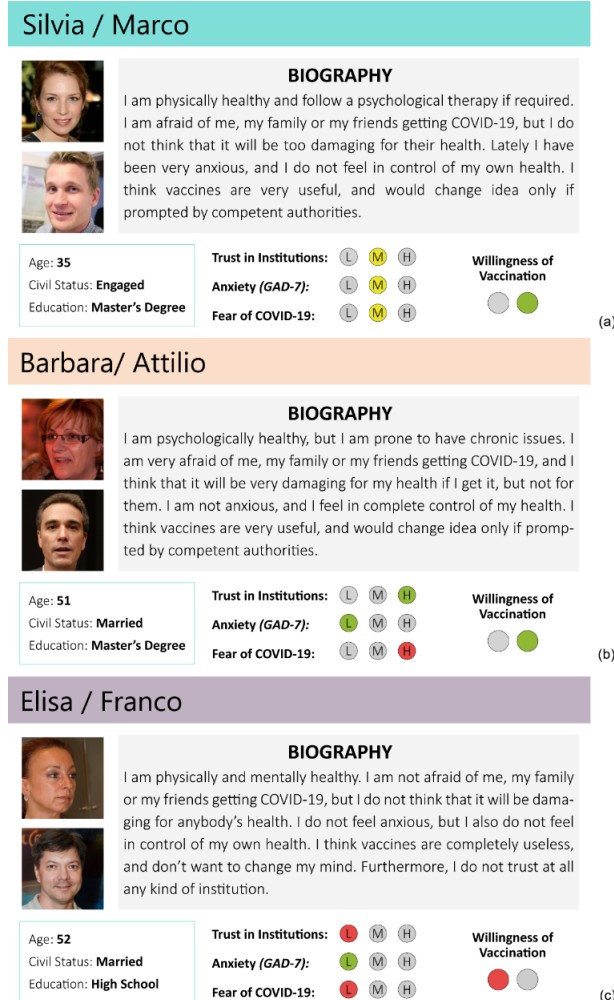

## Silvia / Marco

**BIOGRAPHY**

I am physically healthy and follow a psychological therapy if required. I am afraid of me, my family or my friends getting COVID-19, but I do not think that it will be too damaging for their health. Lately I have been very anxious, and I do not feel in control of my own health. I think vaccines are very useful, and would change idea only if prompted by competent authorities.

Age: 35
Civil Status: **Engaged**
Education: **Master's Degree**

Trust in Institutions: L M H
Anxiety *(GAD-7)*: L M H
Fear of COVID-19: L M H

Willingness of Vaccination

(a)

## Barbara/ Attilio

**BIOGRAPHY**

I am psychologically healthy, but I am prone to have chronic issues. I am very afraid of me, my family or my friends getting COVID-19, and I think that it will be very damaging for my health if I get it, but not for them. I am not anxious, and I feel in complete control of my health. I think vaccines are very useful, and would change idea only if prompted by competent authorities.

Age: 51
Civil Status: **Married**
Education: **Master's Degree**

Trust in Institutions: L M H
Anxiety *(GAD-7)*: L M H
Fear of COVID-19: L M H

Willingness of Vaccination

(b)

## Elisa / Franco

**BIOGRAPHY**

I am physically and mentally healthy. I am not afraid of me, my family or my friends getting COVID-19, but I do not think that it will be damaging for anybody's health. I do not feel anxious, but I also do not feel in control of my own health. I think vaccines are completely useless, and don't want to change my mind. Furthermore, I do not trust at all any kind of institution.

Age: 52
Civil Status: **Married**
Education: **High School**

Trust in Institutions: L M H
Anxiety *(GAD-7)*: L M H
Fear of COVID-19: L M H

Willingness of Vaccination

(c)

Fig. 5. Persona cards as a result of K-Medoids clustering, with K=3 and PCAMIX preprocessing applied to the whole dataset. Silvia and Marco (a), represent cluster 1; Barbara and Attilio (b) represent cluster 2 and Elisa and Franco (c) represent cluster 3.

attributes among each other, in order to be easily distinguishable.

Compared to the total within sum of square distances, the percentage of statistically different attributes provides more precise information, such as a numerical result and a direct comparison between methods. This novel approach overcomes the limitations of the elbow method, which exclusively relies on graphical representations and often requires the input of domain experts.

While the average silhouette gives information on how well separated the clusters are in the k-dimensional space, this is not sufficient for creating effective Personas. In Figure 2, Gower's distance with K=2 clusters and agglomerative hierarchical clustering resulted in the highest average silhouette. However, this would translate in two clusters with n = 1069 for cluster 1 and only n = 1 for cluster 2, not usable for Personas.

The proposed percentage of statistically different attributes provides additional information about the clustering for development of Personas, by shifting the focus from a measure of distance between clusters to what effectively distinguishes Personas. This approach is driven by the recognition that real-world data is highly complex [35], making it challenging to find well-separated clusters of people. Therefore, the primary focus should be on identifying the optimal method capable of correctly separating clusters (with average silhouette > 0), while also providing the highest percentage of statistically different attributes among them and leveraging the elbow method of the total within sum of square differences to solve dubious situations.

This approach not only enhances the robustness of the Persona development process, but also moves towards improved automation and reproducibility. The combination of the novel proposed metric with already established methods creates a more holistic framework for Persona creation, reducing subjectivity and potentially improving the quality and precision of the resulting Personas.

This methodological novelty has implications beyond the specific application to willingness of vaccination against COVID-19, as it can be implemented to Persona development in various fields where understanding the main characteristics of different groups is of crucial importance.

### B. Applicational novelty: Personas for willingness of vaccination against COVID-19 in the general population

This study also represents the first attempt to use Personas to assess the willingness of the general population to receive COVID-19 vaccinations. This approach simplifies the development of interventions by providing a visual representation of the most important characteristics of different groups, that can be used to identify potential tailored interventions to increase vaccine uptake. Persona 3, the one not willing to be vaccinated, presented characteristics that were very different from those of the two other Personas. They were older, and presented lower levels of education, suggesting lower health literacy [36] and potentially facing more difficulties in obtaining and comprehending complex information [37], such as the one related to vaccines. Consequently, information aimed at this Persona should use concise and straightforward language.

Moreover, a correlation was observed between vaccine hesitancy in Persona 3 and mistrust in healthcare and scientific institutions (Spearman's r = 0.296, p-value < 0.001), as well as distrust in vaccine effectiveness (Spearman's r = 0.442, p-value < 0.001) and education. These findings align with existing literature [38], [39], [40], [41], suggesting the necessity of reaching this demographic through diverse sources beyond institutional channels. Given the functional role of social support in promoting behavioral changes [40], information originating from social proximity may be perceived as more trustworthy than institutional sources.

Fear of COVID-19 was also correlated with vaccination willingness (Spearman's r = 0.103, p-value < 0.001), indicating that those with a lower perceived risk of COVID-19 were less inclined to get vaccinated. The study demonstrates that the Persona approach in analyzing vaccination willingness data yields results comparable to established methods such as path analysis and cross-validation [12].

While this study focused specifically on COVID-19 vaccinations, the developed Personas present potential applications beyond this particular context. Most of the goals identified within the Personas, such as trust in institution, anxiety, and sociodemographic factors are not uniquely tied to COVID-19. This suggests that the Personas could be adapted to tailor digital nudges and interventions aimed at increasing

vaccine uptakes for other diseases or public health threats. However, as each disease context may present unique challenges and factors influencing public perception and behavior, further data could be collected to improve the Personas and adapt them to other specific health situations.

### C. Limitations

The main limitation of the presented study is in the numerical disparity between the respondents willing to be vaccinated (85.33%) compared to those that were not (14.67%). This unbalance could introduce difficulties in developing Personas aimed at understanding the characteristics of the minority group. However, these numbers were highly representative of the real-world situation of vaccines in Italy in September 2022, where 86.68% of the general population received at least one dose of COVID-19 vaccination [42]. Furthermore, no validation was performed on the developed Personas, as no golden standard was available.

A limitation of this study is the data collected exclusively in Italy. The COVID-19 pandemic has demonstrated that national responses, healthcare systems, cultural attitudes, and communication strategies can vary widely between countries, potentially influencing vaccination willingness in diverse ways. While some identified factors, such as trust in institutions and fear of the disease, may have broader applicability, we cannot assume that the Personas developed from our Italian dataset would be fully representative of populations in other nations. Future research should aim to collect comparable data from multiple countries to identify both universal and country-specific attributes affecting willingness of vaccination.

## V. Conclusions

In conclusion, this study demonstrated the feasibility of applying a Persona-based approach to understand the characteristics of the population willing, or not, to be vaccinated against COVID-19. By identifying the most relevant attributes of different types of target people, this approach offers a clear and immediate representation of the characteristics of individuals with different willingness to be vaccinated, which can support tailored interventions aimed at increasing vaccine uptake even in potential future contexts.

Moreover, we proposed a novel graph, the percentage of statistically different attributes, to assess the optimal number of Personas to be developed and to choose the appropriate data preprocessing method. Compared to other methods, it provides a more detailed and informative approach for creating effective Personas, facilitating the automation and reproducibility of the Persona development process.

Future work on the methodological implementation should focus on the validation of the obtained Personas, either through external approaches such as scenario development; or internal approaches, focused on data-driven validation. Validation would ensure that the developed Personas are usable in real world context, facilitating an applicational future improvement enabling the developed Personas to be used in the development of tailored digital nudges to increase vaccine perception and uptake in the general population.

## Acknowledgment

The authors acknowledge the support of Dr. Grzegorz Bilo and Dr. Lucia Zanotti of Istituto Auxologico Italiano who provided insights and expertise that greatly assisted the research.

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
