# OpenReview forum: "An improved framework to develop Personas applied to willingness of vaccination in the general population"
_IEEE.org/EMBS/BHI/2024/Conference — IEEE BHI'24_

### Official Review · Reviewer_MHKM · 2024-08-02
**An improved framework to develop Personas applied to willingness of vaccination in the general population**

**Overall Rating:** 6
**Confidence:** 3

**Other Quality Metrics:**

Clarity of writing: great
Clinical Significance: good
Methodological Novelty: fair
Experiments and Results: great.

**Questions For The Authors:**

On page 4, when talking about Figure 4, could you explain why best value for K is 4? The subfigures seem all the same to me.

**Strengths:**

A novel approach is used in clustering, by identifying different characteristics rather than similarities to create Personas.

**Summary Of The Paper:**

This study uses statistical methods to identify the demographic and sociological differences among the groups with vaccine hesitancy.

**Weaknesses:**

Most of the methodology depends on related work including the surveys that were applied. I am not sure there is enough novelty to consider this work for JBHI.

---

### Official Review · Reviewer_mGA3 · 2024-08-03
**Interesting but with flaws**

**Overall Rating:** 6
**Confidence:** 4

**Other Quality Metrics:**

(a) Clarity of writing: Great; (b) Clinical Significance: Good; (c) Methodological Novelty: Fair; (d) Experiments and Results: Good

**Questions For The Authors:**

- Why does the elbow method for PCAMIX indicates K=3? No elbow is observed in Figure 4.
- Why are K=2 and K=3 the optimal values for PCAMIX according to Figure 3? Shouldn't it be only K=2? Please explain.

**Strengths:**

Social relevance of this type of studies which can allow to find proper interventions to increase health literacy.

**Summary Of The Paper:**

The paper presents a methodology to create Personas from a survey related to COVID-19 incidence and willingness to vaccination.

**Weaknesses:**

- The mentioned novel approach of statistical analysis of analysis do not seem novel to me.
- The choice of the number of clusters is not clear
- The choice of the details for the Personas is not well explained. Considering Table 1, what was used to choose the parameters shown in Figure 5?

---

### Official Review · Reviewer_Haie · 2024-08-11
**Improve**

**Overall Rating:** 7
**Confidence:** 5

**Other Quality Metrics:**

(a) Clarity of writing                : GREAT, paper is well-organized and written in a fluent language
(b) Clinical Significance          : GOOD, potential of proposed system for application in clinical scenarios is high, yet additional data are required considering population characteristics and examples from other countries too. Generalization to vaccination against other diseases needs a more detailed and better defined framework
(c) Methodological Novelty     : FAIR, innovative characteristics of presented study are mostly related to the application field and the integration of an alternative indicator, yet further explanation is required. Reproducibility of results by any individual researcher is not feasibility considering only the provided flowchart of the overall data analysis framework
(d) Experiments and Results  : GOOD, since different pre-processing methods are examined and validation is performed on real data. Additional data are required in order to generalize results in international level and other diseases

**Questions For The Authors:**

In order to further improve the quality of the manuscript, the following amendments could be addressed:
* title can be misleading since application of proposed framework is related to COVID-19 pandemic and cannot be generalized in vaccination against several diseases. COVID-19 term should be included
* what is the potential of proposed framework for future application in clinical scenarios since nowadays COVID-19 is considered a “common” annual disease thread and not a pandemic?
* how could proposed framework could be generalized to other diseases (public health threads) too?
* text segments from initial paper template have remained in the manuscript and need to be removed
* what are the specific characteristics introduced by the authors towards improving “version of a framework, present in current literature [9], applied to the development of Personas to assess the characteristics of the population that is willing, or not, to be vaccinated”?
* dealing with COVID-19 pandemic was a complicated procedure since it was a special global situation and decisions taken in national level were quite different, affecting way of thinking of domestic population. How is this parameter addressed in presented study? How are the outcomes of this work intended to be generalized in international level?
* what are the intended future improvements of proposed work?

**Strengths:**

Advantages of presented study can be summarized as follows:
* alternative application of Personas to identify the characteristics of the population that was less willing to be vaccinated against COVID-19
* investigation of different data pre-processing methods for improved accuracy of statistical analysis
* validation on real dataset, consideting all Ethical Issues related to access to clinical information
* investigation of multiple population features within developed statistical analysis framework
* potential to be applied to other diseases

**Summary Of The Paper:**

Presented study introduces an improved version of a framework to create Personas in order to identify the characteristics of the population that was less willing to be vaccinated, with application to COVID-19 pandemic. A new indicator and a comparison of data pre-processing methods are introduced, providing results of increased accuracy and robustness. It is an alternative application of Personas, with potential to identify solutions towards vaccination of general population, based on a wide and indicative dataset created by the authors.

**Weaknesses:**

Weaknesses of presented study can be summarized as follows:
*  it is not clear what the algorithmic modifications of existing reference method are to develop Personas
* reproducibility of results by any individual researcher does not seem to be easy, since no mathematical background and/or algorithmic setup directions are provided. Only the general aspects of data analysis scheme are provided in the process flowchart
* additional data are required in order to generalize outcome of proposed approach, since examined dataset included subjects from Italy and does not cover any special conditions and parameters regarding other countries (e.g. national health systems, mentality of population, actions taken by national authorities, digital and communication means of governments and media)

---

### Decision · Program_Chairs · 2024-09-23

Accept